# Small Angle X-ray Scattering, Molecular Modeling, and Chemometric Studies from a Thrombin-Like (Lmr-47) Enzyme of *Lachesis m. rhombeata* Venom

**DOI:** 10.3390/molecules26133930

**Published:** 2021-06-28

**Authors:** Salvatore Giovanni De-Simone, Guilherme Curty Lechuga, Paloma Napoleão-Pêgo, Larissa Rodrigues Gomes, David William Provance, Vinícius Dias Nirello, Ana Carolina Rennó Sodero, Herbert Leonel de Mattos Guedes

**Affiliations:** 1FIOCRUZ, Center of Technological Development in Health (CDTS), National Institute of Science and Technology for Innovation on Neglected Diseases Population (INCT-IDPN), Rio de Janeiro 21040-900, Brazil; guilherme.lechuga@cdts.fiocruz.br (G.C.L.); paloma.pego@cdts.fiocruz.br (P.N.-P.); larissa.gomes@cdts.fiocruz.br (L.R.G.); bill.provance@cdts.fiocruz.br (D.W.P.J.); 2Department of Cellular and Molecular Biology, Biology Institute, Federal Fluminense University, Niterói 24020-141, Brazil; 3Interdisciplinary Medical Research Laboratory, Oswaldo Cruz Institute/FIOCRUZ, Rio de Janeiro 21040-900, Brazil; herbert@biof.ufrj.br; 4Faculty of Pharmacy, Federal of Rio de Janeiro University, Rio de Janeiro 21949-900, Brazil; vinicius_nirello@id.uff.br (V.D.N.); acrsodero@gmail.com (A.C.R.S.); 5Laboratory of Immunopharmacology, Federal of Rio de Janeiro University, Duque de Caxias 25245-390, Brazil

**Keywords:** *Lachelis m. rhombeata*, serine protease, SVTLE, solution structure, protein shape, guanidine inhibitors, amidine inhibitors

## Abstract

Introduction: Snakebite envenomation is considered a neglected tropical disease, and SVTLEs critical elements are involved in serious coagulopathies that occur on envenoming. Although some enzymes of this group have been structurally investigated, it is essential to characterize other proteins to better understand their unique properties such as the *Lachesis muta rhombeata* 47 kDa (Lmr-47) venom serine protease. Methods: The structure of Lmr-47 was studied in solution, using SAXS, DLS, CD, and in silico by homology modeling. Molecular docking experiments simulated 21 competitive inhibitors. Results: At pH 8.0, Lmr-47 has an Rg of 34.5 ± 0.6 Å, Dmax of 130 Å, and SR of 50 Å, according to DLS data. Kratky plot analysis indicates a rigid shape at pH 8.0. Conversely, the pH variation does not change the center of mass’s intrinsic fluorescence, possibly indicating the absence of fluorescent amino acids in the regions affected by pH variation. CD experiments show a substantially random coiled secondary structure not affected by pH. The low-resolution model of Lmr-47 presented a prolate elongated shape at pH 8.0. Using the 3D structure obtained by molecular modeling, docking experiments identified five good and three suitable competitive inhibitors. Conclusion: Together, our work provided insights into the structure of the Lmr-47 and identified inhibitors that may enhance our understanding of thrombin-like family proteins.

## 1. Introduction

The snake venom has been an essential source of new drugs, especially for bleeding disorders, and even anti-cancer and hemostasis. Moreover, serine proteases have played an important role in some of these disorders [1,2,3]. Some of these venom proteases are often referred to as thrombin-like enzymes (SVTLEs) due to their ability to cleave fibrinogen, releasing fibrinopeptide A, fibrinopeptide B, or both. SVTLEs showed a high similarity within them and a minor degree of similarity with mammalian serine proteases such as trypsin and thrombin [4], and they are potential candidates for antithrombotic therapy. SVTLEs and trypsin-like enzymes have similar primary structures and share a common ancestor [5,6]. SVTLEs have been biochemically well studied, and despite the three-dimensional (3D) structure of saxtrombine [7], AhV TL-I [8], and jararacussin-I [9] by crystallography, there is a lack of 3D structural information and specificity [10] concerning this group of protease. Homology models generated for several SVTLEs have fostered the distinct belief that these proteins would display a globular structure similar to other chymotrypsin family members, e.g., [4]. However, SVTLEs may show significant differences in the molecular shape mainly due to variations in surface loop length and composition [11] and specific carbohydrate content [2].

Previously, we have biochemically characterized Lmr-47, an SVTLE from *Lachesis m. rhombeata* [8], which is capable of cleaving either the α and β chain of fibrinogen, with a preference for the β chain. The enzyme also hydrolyzes synthetic substrates with a specificity similar to trypsin [12] and is competitively inhibited by amidines and guanidines [13].

Structural information on Lmr-47 is essential to characterize this protein further and better comprehend the relationship between the fold, enzymatic activity, inhibitors, and pH dependence. In this context, in this work, we have applied a set of biophysical methods and bioinformatics approaches for understanding the structure and molecular envelope of the Lmr-47 in solution. We also report the effect of pH on Lmr-47 structure by circular dichroism (CD), intrinsic fluorescence (IF), dynamic light scattering (DLS), and small-angle X-ray scattering (SAXS) studies ab initio; models of Lmr-47 were construed from SAXS data at pH 8.0. In addition, we have applied the quantitative structure–activity relationships (QSAR) approach to correlate the biological activities of 21 amidine and guanidine inhibitors with their physicochemical parameters and characterize the enzyme/inhibitor interaction.

## 2. Results

### 2.1. Lmr-47 Structure by DLS and SAXS

Lm-47 was purified from *Lachesis m. rhombeata* venom using two chromatographic steps as described previously [14]. The purified Lmr-47 sample showed a single band by SDS–PAGE analysis under reducing and non-reducing conditions (data not shown). The apparent radius of Lmr-47 was determined at pH 8.0 by DLS and SAXS (Table 1). The molecular mass values calculated from Lmr-47 SAXS data at pH 8.0 were 50 ± 3 kDa and monomeric by SAXS, DLS, and gel filtration chromatography, indicating an absence of aggregation in our experiments.

The superposition of experimental (corrected and normalized) SAXS curves for Lmr-47 at pH 8.0 is shown in Figure 1A. The protein scattering pattern shows a q−4 decay that is compatible with a compact particle. The SAXS profiles (Figure 1A) were collected using low-concentration solutions and using a 1D-gas detector, which does not provide good statistical counting at high angles. For this reason, the curve is noisy in the high q range.

Rg and Dmax values were determined from the pair distribution function, p(r), and obtained an Rg = 37 ± 1 Å and Dmax = 130 Å (Figure 1B). The Rg and the Dmax suggest a prolate elongated shape for pH 8.0. It is noteworthy that the SR value determined by DLS was 50 Å (Figure 1C). These results agree with SAXS data, clearly demonstrating one particle and following the same trend (particle dimension pH 8.0).

### 2.2. Effect of pH by CD and IF

The CD spectrum of Lmr-47 protein in a solution buffered at pH 7.5 is shown in Figure 2A and surprisingly depicted a high content of random coil for an active enzyme belonging to the chymotrypsin fold.

By measuring the IF emission, we were able to determine the global changes in the Lmr-47 at different pH values. Lmr-47 IF had a maximum located at 331.5 nm, with the center of mass at 343 nm, as shown in Figure 2B. No spectral shift in the center of mass was observed by pH modification, which was possibly due to the absence of fluorescent amino acids in the regions affected by pH variation. On the other hand, a variation in fluorescence intensities was observed but rationalized as no simple pH change. It is possible that differences in the protein flexibility at the different pH may be affecting the quantum yields of the fluorophores without changing their exposure to the solvent (as judged by the lack of spectral shift).

### 2.3. Molecular Envelope of Lmr-47

The low-resolution molecular models of Lmr-47 at pH 8.0 were determined using the GASBOR program. Starting from a globular arrangement of interconnected spherical beads (dummy residues) that mimic the protein sequence, the program optimizes the structure by rotation and translation of the dummy residues, using a simulated annealing protocol, to obtain a three-configuration that gives the best fit of the experimental scattering data. The number of dummy residues used in the calculation is provided by the protein size, which is known. Since SAXS is a low-resolution technique, it is impossible to obtain one unique solution from this modeling procedure. To have a more reliable result, we have calculated five independent models, and the data are shown in Figure 3A. The model average was performed by the program package DAMAVER [16]. In the averaging procedure, the calculated models are aligned, compared, and as a result, the most probable configuration is represented by an average model built using a close-packed arrangement of spheres. In agreement with the p(r) function, the Lmr-47 model at pH 8.0 has a prolate shape (Figure 3A). Analysis of the models by Hydropro furnished Rg and Dmax values similar to those obtained from the scattering curves, giving further support to the study of SAXS data.

### 2.4. Molecular Modeling and Dock Experiments

SAR evaluation performed semi-empirical energy minimization calculations (AM1 Hamiltonian) on the serine proteinase inhibitors. The chemical structures of the selected inhibitors are shown in Appendix A. The next step was to choose a set of chemical variables (structural and physicochemical) that describe the target enzyme/inhibitors’ interactions. Previously proposed binding modes [18] determined our selection of structural variables. This model suggests that the electron-donating group present in the inhibitor structure would have a formal positive charge at the central atom of the substituent that attracts the standard negative charge of the OH group of Ser177. Conversely, with electron-withdrawing groups present in the inhibitor, the partial negative charge at the central atom of the substituent should repel the partial negative charge at the OH group of the catalytic site.

The variables dipole moment (µ), charge over selected sites (QD or QA), frontier orbital energies (EHOMO and ELUMO), amino bond length (N-H), and molecular volume can describe reasonably well the interaction site. Molecular volume is directly related to the organization of the inhibitor within the active site pocket. The charge over selected areas (QD and QA) and amino bond length (N-H) are directly related to hydrogen bonds between the inhibitor and protease in the interaction site. The frontier orbital energies (EHOMO and ELUMO) can classify the set of inhibitors in terms of their ability to act as nucleophiles or electrophiles, which is as expected for the type of structure of the inhibitor. Another energetic variable is DH, which classifies the set of inhibitors in relative thermodynamic stability and is expected to group similar inhibitors. Once the chemical variables chosen to relate activity were calculated, all the information that we found necessary was collected, and the data (Appendix A) were converted into a matrix of 21 inhibitors (called objects) and seven variables. Therefore, our working matrix had 21 lines and 7 columns. We found autoscaling necessary because we are working with variables that differ significantly (e.g., DH expressed in kcal mol-1 and EHOMO described in eV), and these differences in magnitude could restrict our analysis. Auto-scaled data were applied as principal components analysis (PCA) and hierarchical clustering analysis (HCA).

A detailed discussion of the theoretical background of principal components analysis (PCA) and hierarchical clustering analysis (HCA) has been supplied elsewhere [19,20]. It is essential to mention that PCA will simplify the objects represented by only a few uncorrelated, new variables (the so-called principal components, PCs), while HCA will show how the objects “cluster” each other in the multivariate space.

The principal components analysis results were quite interesting; we decided to work with the first, second, and third principal components. These three main components respond to 70% of the global variance and describe the aspects of enzyme inhibition in which we are interested.

The chemical variables that significantly influence PCA analysis are charge over selected sites (QA or QD), dipole moment, and electron affinity (ELUMO). These variables reflect the enzyme/inhibitor complexes’ binding mode, based on the charge attraction/repulsion of the inhibitor with the hydroxyl portion of Ser177. The inhibitor’s electron affinity also plays a significant role in the inhibition process, since the LUMO molecular orbital of the inhibitor has to accommodate the partial negative charge of the Ser177 residue in the active site.

Hierarchical clustering analysis (HCA) confirms the results obtained by principal component analysis showing the separation of the set of inhibitors in six distinct groups related to structure and inhibitory activity, as shown in Appendix A. This clustering is made in multivariate space. In other words, the separation is considered with all the seven variables.

Inhibitor 13 is a poor inhibitor and is isolated with only 10% similarity with the rest of the inhibitors set; inhibitors 4 and 19 form a small group, and these are suitable inhibitors with electron-donating groups attached to the structure. Following a group of six poor and moderate inhibitors, 5/8/6/12/9/16, all benzamidine with moderate electron-withdrawing groups were attached. Another two-benzamidine groups are observed: the first containing five aromatic benzamidine with mild to good inhibition profiles (inhibitors 3/10/1/2/7) and the second holding three benzamidine with electron-withdrawing groups attached. These are poor inhibitors (14/15/11). The last group observed contains four guanidine: 17/18/20/21. These are suitable inhibitors, except for guanidine (21), which is the worst inhibitor in this group. 

A 3D structure of Lmr-47 was obtained by homology modeling using a template of the thrombin-like enzyme from *Deinagkistrodon acutus*, which shows 63% of identity. The 3D structure obtained is shown in Figure 3B, with its active site displayed as sticks (Asp88, His43, and Ser177).

Afterward, molecular docking between the serine proteinase and 21 known inhibitors (amidine and guanidine) was conducted (Table 1). The chemical structure of the selected inhibitors is shown in Appendix A. 

Docking results showed compounds **15** and **18** as the most potent inhibitor of the series (Figure 4). They showed hydrogen bond interaction to Cys173, Asn174, Arg175, and Ser177 and a pi–pi exchange to His43 and Trp193 (which composes a hydrophobic pocket wall). Both His43 and Ser177 are catalytic residues, which agree with the presumed active site. The expected repulsion between the inhibitors containing electron-withdrawing groups, as from compound 15, and the aspartic residue in the active site can be reflected as a weaker contact between the inhibitor and the catalytic triad (Asp88, His43, and Ser177) (Figure 4B).

## 3. Discussion

The structures of globular proteins can be classified as native, molten globule, molten globule, and unfolded (random coil) conformations (folding states), according to their secondary structure content and overall flexibility [21]. The molten globule state shows the typical structure of native proteins; however, the molten globule and unfolded conformations do not have rigid tertiary structures.

All known crystallographic structures of the chymotrypsin family members are globular (e.g., thrombin [7,9,22]). As demonstrated in this study, Lmr-47 also has a rigid frame at pH 8.0, as to the SVLT serine protease BJ-46 [11], indicating a possible similarity structural, concerning other members of the chymotrypsin family. Models generated from SAXS data showed that Lmr-47 has a prolate shape and a rigid structure at pH 8.0, as supported by its scattering pattern and its Kratky plot analysis. Since more rigid structures characterize the native state of most proteins compared to folding intermediates [11], Lmr-47 must be in its native form at pH 8.0. Thus, according to the enzyme activity profile for different pH levels, it shows the highest activity at pH 8.0 [10]. Therefore, this finding follows previous observation in another system associated with its anti-coagulant function [23,24]. Similarly, using fluorescence spectroscopy, it was also not feasible to identify an intermediate state, possibly because the tryptophan and tyrosine residues are located in regions unaffected by pH variation. 

Our chemometric analysis shows that PCA and HCA methodologies distinguish the type of the inhibitors. Five good (3/10/1/2/7) and three (17/18/20) suitable inhibitors were identified from a total of 21 evaluated. A separation in terms of the donating or electron-withdrawing groups is clear, corroborating the proposed inhibition mechanism. The adopted model [18] suggests a binding mode in which the electron-donating group present in the inhibitor structure would interact with the formal negative charge of the OH group of Ser177. Our analysis clearly separates inhibitors with electron-withdrawing and donating groups attached to the benzamidine or guanidine core. The separation in terms of inhibitory activity is not as straightforward as in structural features but also shows a slight correlation. Our methodology is capable of grouping the moderate to poor inhibitors better than the suitable inhibitor due to the high similarity in terms of the chemical structure, and another critical point is that we decided not to include the inhibitory effect (KI) as a variable, which affects the separation in terms of inhibitory activity for sure. We chose not to have the KI as a variable in our chemometric analysis because we plan to extend our work to the design of new inhibitors. To apply this methodology to new inhibitors is desirable to work only with calculated variables (KI is an experimental parameter). With this approach, we can analyze the synthesis of the candidate structures.

## 4. Materials and Methods

### 4.1. Lachesis Mutha Rhombeata Venom

*Lachesis m. rhombeata* venom was obtained manually from eight specimens kept at the Vital Brazil Institute (Niterói, Brazil), six received from Alagoas State, and the others in Bahia State. The venom was dried and stored frozen at −20 °C until use. 

### 4.2. Enzyme Purification

Lmr-47 was purified as described before [23]. Briefly, the venom (100 mg) was equilibrat-281 ed in 50 mM Tris–HCl containing 0.5 M NaCl, pH 7.0, and subjected to affinity chromatography on a benzamidine–agarose column. The adsorbed proteins were eluted with 1 283 mM HCl (pH 3.0), containing 0.5 M NaCl. The eluate was collected on ice and immediately concentrated using P10 centrifugal filters. The concentrated post-benzamidine–agarose preparation was injected in HPLC (Shim-pack Diol-150 column, Shimadzu, Kyoto, Japan) previously equilibrated in 50 mM phosphate buffer pH 7.2. The protein was fractionated on an automatic HPLC system (Shimadzu, 10A model, Kyoto, Japan) at a 1 mL/min flow rate for 28 min at 25 °C (Appendix A). The fraction containing Lmr-47 was analyzed by sodium dodecyl sulfate-polyacrylamide gel electrophoresis [24] and mass spectrometry (MS; Appendix A). In this case, the samples were dissolved in 50% (*v/v*) acetonitrile (containing 0.1% (*v/v*) TFA) and analyzed on a QUATTRO II triple quadrupole mass spectrometer (Micromass, Altrincham, UK) equipped with a standard electrospray probe, adjusted to 5 μL/min. The source temperature was maintained at 80 °C and the needle voltage was maintained at 3.6 kV during all experiments, applying a drying gas flow (nitrogen) of 200 L/h and a nebulizer gas flow of 20 L/h. The mass spectrometer was calibrated with intact horse heart myoglobin and its typical cone-voltage-induced fragments.

For SAXS experiments, the purified Lmr-47 sample was dialyzed against 50 mM Tris–298 HCl pH 8.0 and then concentrated to 4 mg/mL using P10 centrifugal filters.

### 4.3. Dynamic Light Scattering (DLS)

DLS analysis was performed with DynaPro Nano star (Wyatt Technology Corp., Santa Barbara, CA, USA) at room temperature (25 °C). About 0.1 mg of Lmr-47 was analyzed for each set of DLS data in 50 mM phosphate buffer, pH 8.0. All protein solutions were filtered through a membrane of 0.2 μm porosity to remove any dust before adding to the sampling cell.

### 4.4. X-ray Scattering Experiments, Equipment Setup, and Data Analysis

Small-angle X-ray scattering (SAXS) experiments were performed at the SAXS beamline of the National synchrotron Laboratory, Campinas, Brazil [25]. The experimental setup included a temperature-controlled, 1.5 mm diameter capillary tube sample holder and a linear position-sensitive detector. Data acquisition was performed by taking five 600 s frames, using a sample detector distance of 446 mm and 849 mm and X-ray wavelength of 1.488 Å which enabled detection of a q range (q = (4π/λ)sin(θ), λ = wavelength and 2θ = scattering angle) equal to 0.0165 Å^−1^ and 0.2192 Å^−1^. All the data treatment of the scattering intensities was performed using the software package TRAT1D [26]. The usual correction for detector homogeneity, incident beam intensity, sample absorption, blank subtraction, and intensities averaging was routine. This software provides the corrected experimental powers and error values. Data analysis and model calculations were performed using GNOM [27] and GASBOR (EMBL, Hamburg, Germany) [28]. Using indirect Fourier transformation, the program GNOM gives the pair distance distribution function p(r) to calculate the experimental radius of gyration (Rg) and maximum particle dimension (Dmax). The program GASBOR was used to generate ab initio models as described before. To calculate the hydrodynamic parameters, radii of gyration, and maximum distances from the ab initio models, enabling direct comparison with the experimental values, we used the HydroPro software [29].

### 4.5. Circular Dichroism Analysis

For CD measurements, a Jasco J-810 spectro-polarimeter was used. Lmr-47 was analyzed at 50 mM phosphate buffer pH 6.0 and 10 mM Tris–HCl pH 7.5 and 8.0. The data were collected using a scanning rate of 50 nm/min with a spectral bandwidth of 1 nm using a 0.2 cm path-length cell in far-UV (195–250 nm). Measurements were carried out at room temperature (25 °C) at a final protein concentration of 1 mM. All buffers used were of analytical grade and were filtered before use to avoid light scattering by small particles.

### 4.6. Intrinsic Fluorescence (IF)

IF measurements were followed by exciting the protein samples at 280 nm and measuring the fluorescence emission between 300 and 400 nm. For IF, the center of spectral mass bνN quantified the spectra according to the equation below [30]:bmN ¼mi Fi = Fi.

Fi is the fluorescence emitted at wave number νi, and the summation is carried out over the range cited above. Experiments were performed at 25 °C using 100 mM sodium citrate (pH 5.0 and 5.5), 100 mM phosphate buffer (pH 6.0 and 6.5), and 100 mM Tris–HCl (pH 7.0–9.0). The experiments were performed on a JASCO FP-6300 fluorescence spectrometer (Jasco, Inc., Easton, PA, USA).

### 4.7. Protein Determination

The measurement of protein concentration was conducted using the DC protein assay (Bio-Rad, Hercules, CA, USA). Bovine serum albumin was used as standard.

### 4.8. Molecular Modeling

The model was built using the Modeller 10.1 program [31], which uses the spatial constraints satisfaction method. A hundred models were built, among which a final model chose a final model based on the parameter function of DOPE (Discrete Optimized Protein Energy). The crystallographic coordinates used as a template for the homology modeling were thrombin-like enzymes from *Deinagkistrodon acutus* (PDB code 5XRF). 

The evaluation of the stereochemical quality of the models was performed using the PROCHECK v3 [32] and VERIFY-3D [33,34] online servers and was applied to the default settings of the two servers mentioned.

We used the AutoDock Vina v1.1.2 program [35] to perform docking assays. The protonation states of the model were analyzed at pH 8.0 by the PROPKA server, and the addition of hydrogens, followed by the calculation of the Gasteiger load, were performed by AutoDockTools4. Grids with 15 × 15 × 15 were used, centered on the midpoint of the catalytic triad, and considered the standard spacing between the grid points (0.375 Å). This approach generated nine poses per ligand. All poses obtained were submitted to re-evaluation by the Open Drug Discovery Toolkit (ODDT) [36] software, using the Random Forest-based v3 scoring function (RF-Score v3) [37].

Edition and preliminary minimization of inhibitors were done with Windows v5.1 (Serena Software) using the MMX force field. The inhibitors were modeled using the semi-empirical MOPAC v7.0 package (AM1 Hamiltonian) to obtain the desired structural and physicochemical parameters. The chemometric analysis was conducted using the ARTHUR/UNICAMP public domain package. The programs used were ENTER for the edition of the training set, SCALE for the autoscaling, KARLOV for principal component analysis, and HIER for hierarchical clustering analysis. 

## 5. Conclusions

The present study demonstrated that the Lmr-47 has a prolate shape in its native state (pH 8). The PCA and the HCA methods proved their ability to help extract relevant information on the quantitative structure–activity relationship for benzamidine and guanidine inhibitors of the *Lachesis muta rhombeata* serine proteinase. The PCA method gave interesting results identifying the significant chemical variables related to the inhibitory activity. These variables reflect the binding mode of the protease–inhibitor complexes. Molecular docking simulations also revealed details of the interaction site essential to selecting the chemical variables chosen to relate inhibitory activity.

The detailed analysis of all the possible protease–inhibitor interactions is a difficult task, and when comparing several chemical variables, the evaluation of the quantitative importance of different effects is sometimes subjective. The PCA method permitted us to focus our attention on selected chemical variables and provided objective information on their relative importance, which will use the information to design novel inhibitors. Hierarchical clustering analysis showed an interesting correlation between the inhibitors’ chemical structure and, less clearly, their inhibitory activity. Our methodology has been developed for practical purposes, and the results can be directly applicable.

## Figures and Tables

**Figure 1 molecules-26-03930-f001:**
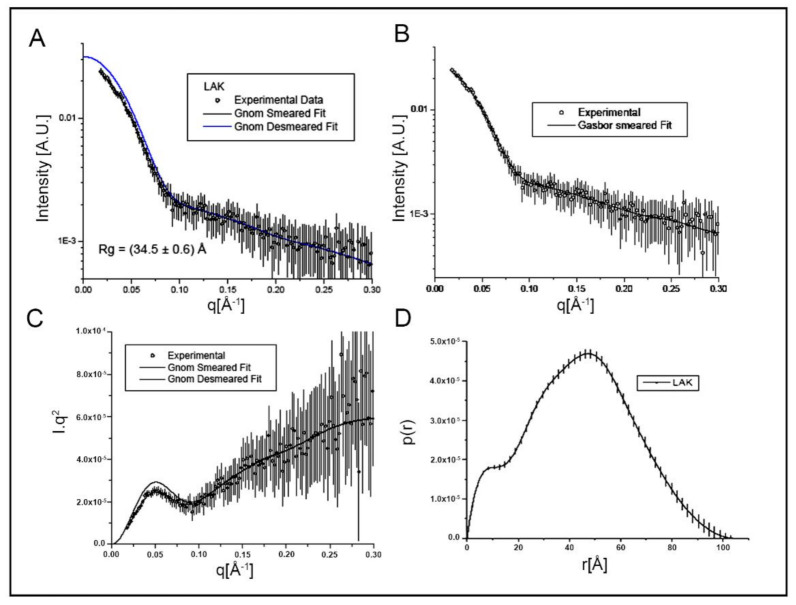
SAXS analysis of Lmr-47 structure, flexibility (**A**–**C**), and Kratky plot at pH 8.0 (**D**). Small-angle scattering pattern (**A**–**C**) and pair distribution function [p(r)] (**D**). The p(r) function and fitted curves were calculated from experimental scattering data using the program GNOM [15]. For details, see text. The curves for thermally denatured Lmr-47 (without enzymatic activity) at pH 8.0 are plotted for reference.

**Figure 2 molecules-26-03930-f002:**
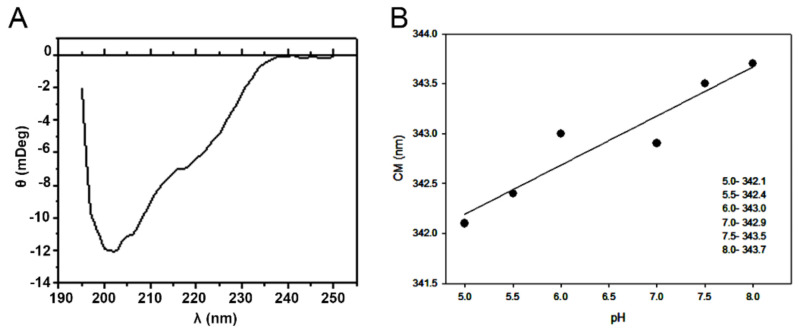
Circular dichroism analysis of the Lmr-47 (**A**) and intrinsic fluorescence spectra on varying pH (**B**). CD spectra of Lmr-47 at 1 mg/mL were recorded at pH 7.5 in 50 mM Tris–HCl at 25 °C. Experiments of IF were performed in 100 mM sodium citrate (pH 5.0 and 5.5), 100 mM phosphate buffer (pH 6.0 and 6.5), and 100 mM Tris–HCl (pH 7.0–8.0) with 0.2 mg/mL of Lmr-47 at 25 °C. Samples were excited at 280 nm, and fluorescence emission was measured between 300 and 400 nm.

**Figure 3 molecules-26-03930-f003:**
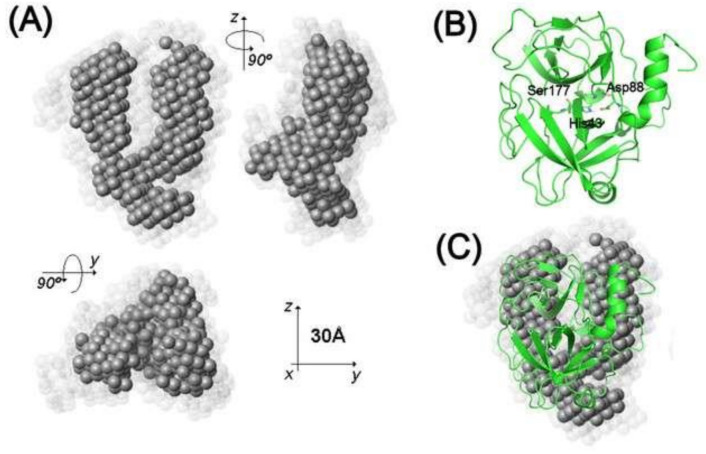
Molecular envelopes, tridimensional structure, and docking of Lmr-47. (**A**) Molecular envelope determined by SAXS at pH 8.0. Ab initio models represent an average of five dummy residue models (DRM) calculations using GASBOR [17]. The light gray spheres in (**A**) represent the space accessible for the DRM as judged by the degree of conservation of a given dummy residue among the five models. (**B**) The three-dimensional structure was obtained by comparative modeling. The structure is presented as a green ribbon, with its active site displayed as sticks (Asp88, His43, and Ser177). (**C**) Superposition of SAXS molecular envelope and the model built by comparative modeling. (**C**) Selected resulting structures of docking calculations with the 4-aminophenylguanidine complex, with the inhibitor colored in blue and the protease in green.

**Figure 4 molecules-26-03930-f004:**
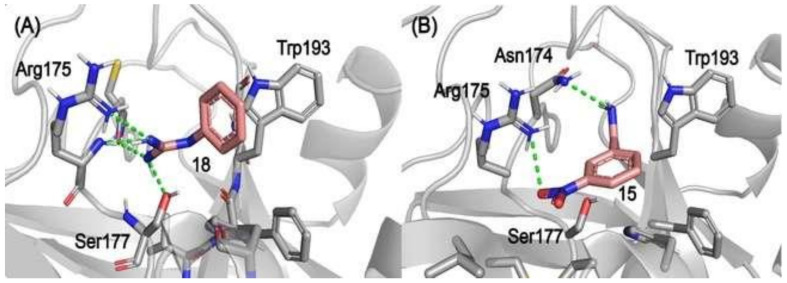
Molecular docking results of inhibitors **18** (**A**) and **15** (**B**) into the Lmr-47 binding site. The protein structure is presented as a gray ribbon, with its residues as sticks; inhibitors are colored as salmon carbons. Hydrogen bonds are represented as dashed green lines.

**Table 1 molecules-26-03930-t001:** Results from molecular docking of inhibitors and *Lmr-47*.

Inhibitor	Vina_Score (kcal/mol)	RFScore_v3
1	−4.5	4.4
2	−3.9	4.4
3	−3.9	4.3
4	−4.4	4.1
5	−4.4	3.9
6	−4.1	3.9
7	−4	4.2
8	−4.1	3.9
9	−4.1	3.8
10	−4.2	4.6
11	−4.2	4.7
12	−4.0	4.1
13	−4.3	4.8
14	−4.1	4.9
15	−4.5	5.2
16	−3.7	4.5
17	−4.2	4.7
18	−4.2	5.0
19	−4.4	4.9
20	−3.4	4.3
21	−3.3	4.4

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
