# Peer review of "Small Angle X-ray Scattering, Molecular Modeling, and Chemometric Studies from a Thrombin-Like (Lmr-47) Enzyme of Lachesis m. rhombeata Venom"

_molecules, 2021, doi:10.3390/molecules26133930_

Round 1

Reviewer 1 Report

The manuscript "Conformational states, molecular modeling, and chemometric studies on a thrombin-like (Lmr-47) enzyme from Lachesis m. rhombeata venom" by De-Smone et al. is about the structural determination of snake venom Lmr-47 enzyme using SAXS, DLS, CD, and molecular modeling techniques. Also, docking and chemometrics studies for 21 competitive inhibitors are included in the manuscript. The results presented in the manuscript are meaningful but the presentation is rather poor and must be improved before accepting this manuscript for publication.

Major remarks:

  1. Line 138 – (Table 1) – there is no Table 1 in the manuscript.
  2. Molecular models of Lmr-47 were determined with two methods: from SAXS data (using GASBOR program) and from homology modeling, but the authors didn’t correlate two models. What are the Rg and Dmax values for models obtained by homology modeling? Can the two models be overlapped?
  3. Results of the docking study are not shown. The authors call some inhibitors poor, some moderate, some good, and some suitable, but I don’t see criteria by which they decide the inhibitor strength. At least the scoring function from the docking experiment should be presented. Also, some analysis of protein-ligand interactions at least for representatives of poor, suitable, and good inhibitors would make the manuscript more interesting.
  4. In the PCA and HSA study, only some intrinsic inhibitor variables calculated with the semiempirical AM1 method are included. First, I think that the applied method is very low and that some higher-level theory methods (for example DFT) should be used. Second, I don’t understand some variables listed in Table S1. What does Q represent in compounds with two electron donor or electron acceptor groups? For example, in Compound 1 there is only an amidine group (so I guess Q is the total charge of this group), but in compound 2 there are amidine group and amino group at the para position of the benzene ring. To which of these two groups is Q in table S1 related. The same goes for N-H bond length (is it amidine or amino N-H in compound 2?). Also, what DH represents? Is it the enthalpy of formation? Please be specific.
  5. The sentence (Lines 157-159) “This model was similar to that obtained by the I-TASSER (C-score 1.02, TM-score=0.85 ± 0.08; RMSD=3,6 ± 25 Å) server using 90 templates (data not shown).” should be removed from the manuscript. Since you didn’t show us the data this statement is not verifiable.
  6. Line 120 Figur 2 caption – “Fluorescence intensity is reported in arbitrary units (a.u.)”. Florescence results are not shown in the manuscript (at least I couldn’t find them) so it is irrelevant in which units are they not shown.
  7. The resolution of Supplementary Figure 1 is very low, I couldn’t read the substituents for some of the compounds.
  8. Technically manuscript needs to be improved. The Greek letters (μ-for dipole moment, D, etc) needs to be inserted. Also, IF center of mass equation (Line 331) needs to be presented separately.
  9. English is rather poor, some language editing is needed.
  10. Finally, the “Conformational states” part title should be changed – I didn’t found any discussion about different conformations of Lmr-47 enzyme.

Generally, the manuscript needs much improvement in the presentation of the obtained results, so I must recommend MAJOR REVISION for publication in Molecules journal.

Author Response

Major remarks:

  1. Line 138 - (Table 1) - there is no Table 1 in the manuscript.

R: This was unintentional. We have included the missing table.

  1. Molecular models of Lmr-47 were determined with two methods: from SAXS data (using GASBOR program) and from homology modeling, but the authors didn’t correlate two models. What are the Rg and Dmax values ​​for models obtained by homology modeling? Can the two models be overlapped?

R: The modeling experiments were redone and the models were overlapped as suggested (please see Fig. 3)

  1. Results of the docking study are not shown. The authors call some inhibitors poor, some moderate, some good, and some suitable, but I don’t see criteria by which they decide the inhibitor strength. At least the scoring function from the docking experiment should be presented. Also, some analysis of protein-ligand interactions at least for representatives of poor, suitable, and good inhibitors would make the manuscript more interesting.

R: We appreciate the suggestion and add the inhibitors table of docking results and a figure of 3D representation of docking results obtained to inhibitors 15 and 18.

  1. In the PCA and HSA study, only some intrinsic inhibitor variables calculated with the semiempirical AM1 method are included. First, I think that the applied method is very low and that some higher-level theory methods (for example DFT) should be used. Second, I don’t understand some variables listed in Table S1. What does Q represent in compounds with two electron donor or electron acceptor groups? For example, in Compound 1 there is only an amidine group (so I guess Q is the total charge of this group), but in compound 2 there are amidine group and amino group at the for position of the benzene ring. To which of these two groups is Q in table S1 related. The same goes for N-H bond length (is it amidine or amino N-H in compound 2?). Also, what DH represents? Is it the enthalpy of formation? Please be specific.

R: The molecular modelling studies were improved and all of these questions discuted.

  1. The sentence (Lines 157-159) “This model was similar to that obtained by the I-TASSER (C-score 1.02, TM-score = 0.85 ± 0.08; RMSD = 3.6 ± 25 Å) server using 90 templates (data not shown). ” should be removed from the manuscript. Since you didn’t show us the data this statement is not verifiable.

R: We are in according, this information was removed

  1. Line 120 Figure 2 caption - “Fluorescence intensity is reported in arbitrary units (a.u.)”. Florescence results are not shown in the manuscript (at least I couldn’t find them) so it is irrelevant in which units are they not shown.

R: The results of fluorescence are shown in the Fig 2, and the units removed (please see lines 109-116.

  1. 7. The resolution of Supplementary Figure 1 is very low, I couldn’t read the substituents for some of the compounds.

R: The size of the structures was adequate to the space available in the magazine and a higher resolution of the figure was obtained (Supplementary Fig 3).

  1. Technically manuscript needs to be improved. The Greek letters (μ-for dipole moment, D, etc.) needs to be inserted. Also, IF center of mass equation (Line 331) needs to be presented separately.

R: We have reviewed the manuscript to correct any typing technical errors.

  1. English is rather poor, some language editing is needed.

R: We have had the manuscript reviewed by a native English speaker.

  1. Finally, the “Conformational states” part title should be changed - I didn’t found any discussion about different conformations of Lmr-47 enzyme.

R: The header and text have been revised to rectify this observation. Thank you.

Reviewer 2 Report

The manuscript reports a study of Lmr-47, a thrombin-like enzyme by different empirical (CD, IF, DLS and SAXS) and in silico (homology modelling, docking, PCA and QSAR) methods aimed at understanding its molecular structure and shape in solution.

In principle, such a characterisation by biophysical and bioinformatics methods would lead to a good description a protein.  Unfortunately, the reported results are very sparse; the data necessary to answer the questions asked by the authors is completely lacking.

Since the shape of Lmr-47 is ‘elongated’ (by SAXS) and its structure showed ‘a high content of random coil’ (by CD) which strongly distinguishes this protein from known structures of other chymotrypsins, analysis of this protein should be done very carefully.

First, the SDS-page of Lmr-47 purified under reducing and non-reducing conditions should be presented either in the manuscript or in the supplementary materials. It will be preferable to assess the purity of the proteins with MALDI-TOF mass spectrometry.

Second, to obtain a credible model, the very rough homology pro-model must be refined by the extended molecular dynamics simulation to rich the well equilibrated conformations which can further be used for docking trials or analysis of the protein features.

Remarques:

  1. Line 71: Small angle x-ray scattering is usually abbreviated as SAXS.
  2. Table 1 is missing.
  3. Rg is reported as 34.5 ± 0.6 Å (in Figure 1A) and 37 ± 1 Å in the text.
  4. The effect of pH by IF emission (pH 5.0 to 8.0) is not explained.
  5. A homology model is reported without any detail, without the sequences of the template and the target, without a description of the structure (shape, fold). The presented figure of Lmr-47 is very approximate and shown in a single projection. Such a presentation is not sufficient even for form evaluation, let alone discussing structural features.
  6. How is it possible to tell a similarity if the models (homology and predicted by I-Tasser) showed an RMSD of 3.6 ± 25 Å?
  7. No mooring results, scores, poses. Only the presence of the single inhibitor in the active site is visualized without any detail.
  8. PCA: no results are presented.

Author Response

-Since the shape of Lmr-47 is 'elongated' (by SAXS) and its structure showed 'a high content of random coil' (by CD) which strongly distinguishes this protein from known structures of other chymotrypsins, analysis of this protein should be done very carefully.First, the SDS-page of Lmr-47 purified under reducing and non-reducing conditions should be presented either in the manuscript or in the supplementary materials.

R: To meet the reference, we introduced purification analyzes (Supplementary Figure 1).

It will be preferable to assess the purity of the proteins with MALDI-TOF mass spectrometry. Second, to obtain a credible model, the very rough homology pro-model must be refined by the extended molecular dynamics simulation to rich the well equilibrated conformations which can further be used for docking trials or analysis of the protein features.

R: A figure with the results of the mass spectrometry was included (please see Fig 2) and a new study of docking and refinement was carried out and introduced in the text and discussion (the major modifications are highlighted in yellow).

Notes:

  1. Line 71: Small angle x-ray scattering is usually abbreviated as SAXS.

R: We have changed the abbreviation.

  1. Table 1 is missing.

R: This table does not exist and the information has been removed from the text.

  1. Rg is reported as 34.5 ± 0.6 Å (in Figure 1A) and 37 ± 1 Å in the text.

R: We have corrected this error.

  1. The effect of pH by IF emission (pH 5.0 to 8.0) is not explained.

R: This information has now been entered.

  1. A homology model is reported without any detail, without the sequences of the template and the target, without a description of the structure (shape, fold). The presented figure of Lmr-47 is very approximate and shown in a single projection. Such a presentation is not sufficient even for form evaluation, let alone discussing structural features.

R: Molecular modeling studies using another approach have been redone and new figures and information introduced throughout the text. Thank you.

  1. How is it possible to tell a similarity if the models (homology and predicted by I-Tasser) showed an RMSD of 3.6 ± 25 Å?

R: Homology model was rebuild by Modeller program and I-Tasser model were removed from analysis.

  1. No mooring results, scores, poses. Only the presence of the single inhibitor in the active site is visualized without any detail.

R: We appreciate the suggestion and add the inhibitors table of docking results and a figure of 3D representation of docking results obtained to inhibitors 15 and 18.

8.PCA: no results are presented

R: New analyzes were made with other software and the results included in the text including those of the PCA.

Reviewer 3 Report

In the manuscript titled “Conformational states, molecular modeling, and chemometric studies on a thrombin-like (Lmr-47) enzyme from Lachesis m.rhombeata venom” the structure of Lmr-47 was studied using experimental (SAXS, DLS, CD) and in silico methods. Overall, the work is well designed and executed.  The research study presented in this manuscript is important and exciting. I have one question/suggestion:

  • Homology modeling of Lmr-47 is not validated. It is very important to validate the homology model structure before you perform molecular docking studies.
  • Molecular docking results are not well presented. I don’t see the detailed information regarding the protein-ligand interaction and its importance.
  • Figure 3 docking figures are not clearly labeled. Retake the docking figures.

Author Response

  • Homology modeling of Lmr-47 is not validated. It is very important to validate the homology model structure before you perform molecular docking studies.
  • Molecular docking results are not well presented. I don’t see the detailed information regarding the protein-ligand interaction and its importance.
  • Figure 3 docking figures are not clearly labeled. Retake the docking figures.

R: Molecular modeling studies and docking using another approach have been redone and new figures and information introduced throughout the text. The homology model was rebuilt by the Modeller program and the I-Tasser model was removed from the analysis. We appreciate the suggestion and add the inhibitors table of docking results and a figure of 3D representation of docking results obtained to inhibitors 15 and 18.

Round 2

Reviewer 1 Report

The Manuscript now can be published in Molecules journal. 

Reviewer 2 Report

The manuscript has improved considerably and this revised form can be accepted for publication.